# DESIGNING CONTRACTS FOR EFFORT AND REGULATORY COMPLIANCE

## ABSTRACT

We study mechanism design for principals who must both incentivize agents to exert high-quality effort and ensure that their actions remain strictly legal and compliant. In certain two-sided markets, agents' actions are largely unobservable, and they may face significant extra costs to maintain compliance. Agents maximize their own utility, while principals seek mechanisms that induce both effort and adherence to rules. We propose a contract framework based on hypothesis testing that combines payments with random inspections to deter strategic misbehavior and promote legal, high-quality actions.

## 1 INTRODUCTION

**Motivation.** Principals often work with agents whose actions are only partly observable. Two main frictions create inefficiency. First, **quality**: agents may invest too little in actions that improve the true quality of a product or task. Second, **compliance**: agents may misreport or manipulate evidence to obtain higher payoffs. Classical tools treat these issues separately. Statistical hypothesis tests can screen for quality, but they are vulnerable to selective design, data dredging, or report manipulation. Inspections can deter illegal behavior, yet they are costly and hard to scale. What is missing is a single mechanism that integrates both tools, combining statistical testing with random inspections so that high effort and truthful conduct remain incentive compatible under realistic cost limits. A well-known example is the 2004 Vioxx case, in which Merck withdrew a painkiller after studies revealed elevated cardiovascular risks (Topol (2004)). The episode led to thousands of lawsuits and multi-billion-dollar settlements, becoming a landmark failure in drug safety oversight.

**Example.** The U.S. Food and Drug Administration (FDA) evaluates drug efficacy mainly through clinical trials analyzed with statistical tests. However, statistical evidence alone is vulnerable to strategic behavior such as outcome switching or irregularities at trial sites. In practice, the FDA supplements hypothesis testing with audits and inspections of trial sites, data, and manufacturing. These inspections are costly, but they serve as powerful tools to ensure accurate reporting.

**Challenge.** The example above illustrates the design challenge: how can a principal jointly select a Statistical Contract, which specifies cutoffs, scoring rules, and the mapping from evidence to rewards, and an Inspection Policy, which determines whom to inspect and with what probability, so that agents are motivated to provide high-quality effort while staying compliant? Although adverse selection and compliance are well studied, earlier work often treats testing and inspections separately. Contracts that rely on statistical evidence show how tests can be built into incentives to screen for high-quality types Dütting et al. (2019). However, these approaches do not directly deter illegal behavior when the evidence itself can be manipulated. Inspection-based contracts, by contrast, justify limited random audits as costly deterrents Jost (1991; 1996). In short, hypothesis testing alone cannot ensure compliance, and inspections alone are costly and provide limited information.

**Contributions.** Previous studies usually design mechanisms with a single objective, either to encourage agents to follow safety standards or to motivate them to choose high-quality actions. In contrast, the mechanism proposed in this paper achieves both goals at the same time. Our main contributions are summarized below:

- **Methodologically,** we present a contract design framework that combines statistical inference with contract theory. This framework allows the principal to align agent incentives

using statistical evidence without needing to know the distribution of agent types. We introduce an incentive-compatible contract with a minimum inspection probability to deter unsafe behavior.

- **Theoretically,** we incorporate e-values into contract construction and prove that inspections strengthen incentive effects under Building on earlier work in menu design (for example, Bates et al. (2024); Guruganesh et al. (2023)). Our results show that contracts and inspections work together as complementary tools to ensure agent compliance.

- **Experimentally,** we conduct extensive simulations to evaluate the proposed mechanism under a variety of agent cost structures, inspection budgets, and data distributions. The experiments compare our approach with leading baselines, including the method in Fallah & Jordan (2023); Bates et al. (2024). Results show that our mechanism consistently induces higher quality effort and stronger compliance while maintaining lower total cost. We also perform sensitivity analyses to test robustness when model assumptions or parameter settings change.

### 1.1 RELATED WORK

We consider related research on both contract theory and principal-agent models with hidden actions.

**Contract Theory.** Recent work applies tools from algorithmic game theory to contract theory, with an emphasis on statistical inference and computation. Studies such as Dutting et al. (2021), Guruganesh et al. (2020), and Castiglioni et al. (2021) explore how simple contracts can approximate optimal efficiency. For example, Dütting et al. (2019) shows that in the model of Carroll (2015), an optimal linear contract performs within a constant factor of the best possible contract. Other research integrates contract theory with statistical inference. Papers including Schorfheide & Wolpin (2012), Schorfheide & Wolpin (2013), and Spiess (2018) study inference problems for strategic agents, while Frazier et al. (2014) investigates reward learning agents in experimental environments. In a regulatory setting closer to ours, Min (2023) proposes a model in which firms of different sizes choose between low-cost and high-cost trials. We share the view of the principal as an incentive-driven regulator, but we focus on how the regulator can design statistical inference that remains incentive compatible.

**Mechanism Design.** Our model fits within principal-agent frameworks with hidden actions and connects to several strands of mechanism design. The closest work examines random monitoring in contract design, including Jost (1991), Jost (1996), Strausz (1997), and Barbos (2022). We extend this line by introducing a partial inspection model in which inspections can fully verify compliance with safety standards. Our study also relates to the literature on costly inspection or verification, such as Ben-Porath et al. (2014), Mylovanov & Zapechelnyuk (2017), and Li (2020), and to work on mechanism design with partial or probabilistic verification, including Green & Laffont (1986), Ball & Kattwinkel (2019), Caragiannis et al. (2012), and Ferraioli & Ventre (2018). Inspections in our model are explicitly costly and interact with a statistical contract that discourages data fabrication and preserves the validity of evidence under information asymmetry.

## 2 PRELIMINARIES

We study a principal who contracts with agents who differ in effort (quality) and honesty. The principal uses two instruments: i) a menu of contracts $F(\cdot)$ tied to statistical evidence, and ii) an inspection policy $\beta(\cdot)$ that specifies the probability of auditing reported evidence. The principal's objective is to induce high effort and truthful (compliant) behavior given limited inspection resources.

At each interaction the agent chooses an effort level $a$ (with cost $c(a)$) and a compliance choice $b$, where $a \in \mathcal{A} = \{a_1, \ldots, a_m\}$ and $b \in \{0, 1\}$. Here, $b = 1$ denotes a legal (compliant) action and $b = 0$ denotes an illegal or manipulative action. The agent's action $(a, b)$ is not directly observable to the principal. Instead, the interaction generates an evidence signal $Z$, drawn according to a distribution that depends on $(a, b)$. The principal offers a menu $F(Z)$ mapping observed evidence to payments or assignments, and chooses an inspection probability $\beta(Z)$. The agent's payoff equals the payment from $F$ minus the cost of effort $c(a)$ (payoff is 0 when an audit detects non-compliance). In contrast, the principal's payoff equals the value of the true quality produced minus the costs

of payments and inspections. For exposition, we analyze a single representative agent $i$.[1] For generality, we assume that the effort level $a$ is drawn from a distribution $Q$ over $\mathcal{A}$. Different choices of $Q$ affect quantitative predictions in numerical examples but do not change the qualitative structure of the optimal contracts; we specify $Q$ in Section 4 for numerical evaluation.

To keep the analysis tractable, we impose three standard assumptions. i) **Evidence monotonicity**. The evidence distribution satisfies the monotone likelihood ratio property (MLRP), so higher effort shifts the distribution toward stronger evidence, enabling incentive-compatible screening. ii) **Convex effort cost**. The cost function $c(a)$ is convex, ensuring the existence and uniqueness of the agent's optimal effort and ruling out corner solutions. iii) **Limited budget**. The principal cannot audit all agents at all times, reflecting realistic resource constraints.

Choosing effort $a_k$ incurs cost $c(a_k) \geq 0$, and the principal's expected payoff from the corresponding true quality is $r(a_k) \in \mathcal{R}$. We normalize the effort scale so that $0 = a_1 \leq a_2 \leq \cdots \leq a_m = 1$ and impose the following monotonicity condition.

**Assumption 1** (Monotone costs and benefits). *Costs and payoffs are non-decreasing in effort:*

$$0 = c(a_1) \leq c(a_2) \leq \cdots \leq c(a_m), \qquad 0 = r(a_1) \leq r(a_2) \leq \cdots \leq r(a_m).$$

To credibly demonstrate quality, an agent may produce evidence $Z$ through a compliant process (e.g., clinical trials, standardized tests, proctored assessments). This process imposes a compliance or testing cost $d_t \geq 0$ on the agent. A strategic agent may instead take non-compliant actions ($b = 0$) to avoid this cost and produce manipulated or low-fidelity evidence at negligible cost.[2]

We model the observed evidence as the sum of two components:

$$Z = Z(a, b) = Z_a + Z_b,$$

where $Z_a \sim Q_a$ is the effort-driven component and $Z_b$ is an honesty-driven component. We assume $Z_b \geq 0$ for all realizations, with $Z_b = 0$ when $b = 1$ (compliant/legal action) and $Z_b \geq 0$ when $b = 0$ (non-compliant/illegal action). Under this decomposition, higher effort shifts the distribution of $Z_a$ toward stronger evidence (we assume MLRP for $Q_a$), while dishonest actions add a non-negative bias to observed evidence.

Each contract between the principal and an agent consists of two key design elements: i) a menu of payment functions $F = \{f_j\}_{j=1}^l$, and ii) an inspection policy with probability $\beta$. After observing the evidence $Z$ submitted by the agent, the principal offers the menu $F$. The agent selects the payment function $f_j$ that yields the highest payoff. In the context of drug approval, for example, $F$ represents $l$ possible regulatory pathways, each prescribing a payment $f_j$. The principal inspects the agent with probability $\beta$ and incurs a cost $d_k$ for inspections. An inspection reveals whether the agent complied with safety requirements $b = 1$. Agents who pass inspection receive payment $f_j(Z)$, and agents found non-compliant receive no payment.

---

**Statistical Contract**

The agent chooses whether to participate or withdraw under the following conditions:
1. The agent chooses compliance choice $b \in \{0, 1\}$. If $b = 1$, the agent incurs a compliance cost $d_t \geq 0$.
2. The agent selects a payment function $f$ from the menu $F$.
3. Evidence is produced as $Z = Z_a + Z_b$, with $Z_a \sim Q_a$ and distortion $Z_b \geq 0$.
4. The agent is inspected with probability $\beta$ and, if $b = 0$, triggers a negative effect with independent probability $\alpha$.
5. The agent receives payment
$$T = \begin{cases} f(Z), & b = 1; \\ (1-\alpha)(1-\beta)f(Z), & b = 0. \end{cases}$$

---

If an agent acts illegally ($b = 0$), there is an independent probability $\alpha$ that a negative outcome occurs. Exposure of such an outcome causes a catastrophic reputational loss for the principal. For instance, the collapse of public trust in a regulator such as the FDA or the destruction of a firm's

---

[1]In practice, the principal interacts with $n$ agents; we focus on agent $i$ for exposition, and all symbols henceforth carry the subscript $i$.

[2]Inspections and the statistical contracts introduced in Section 3 deter such behavior.

brand. We assume the events of inspection and negative effects are independent. We capture this by assigning the principal a utility of $-\infty$ when the agent's illegal actions remain undiscovered by the principal and result in catastrophic consequences. This follows earlier work that treats reputational failures as absorbing catastrophic events Fallah & Jordan (2023). The principal first announces the statistical contract, after which the agent decides on participation, effort, and compliance, generates evidence, and is potentially inspected.

## 3 CONTRACTS DESIGN

We now take the principal's perspective and design a mechanism that induces both **high quality** and **legal compliance**. Agent behavior can be classified along two dimensions, quality ($A_l$ vs. $A_h$) and compliance ($b = 0$ vs. $b = 1$), yielding the four cases in Table 1. The action space $\mathcal{A}$ is partitioned into $A_l$ (low quality, e.g., ineffective drugs) and $A_h$ (high quality, e.g., effective drugs), with $A_l \cap A_h = \emptyset$ and $A_l \cup A_h = \mathcal{A}$. The principal's target is $(A_h, b = 1)$, which is high-quality and fully compliant actions.

Table 1: Possible agent behaviors.

|  | $b = 0$ (illegal) | $b = 1$ (legal) |
|---|---|---|
| $A_l$ (low quality) | $(A_l, b = 0)$ | $(A_l, b = 1)$ |
| $A_h$ (high quality) | $(A_h, b = 0)$ | $(A_h, b = 1)$ |

The agent's expected utility for action $(a, b)$ is

$$U_{agent}(a,b) = \begin{cases} f(Z) - c(a) - d_t, & b = 1; \\ (1-\alpha)(1-\beta)f(Z) - c(a), & b = 0. \end{cases} \tag{1}$$

If an agent takes an illegal action ($b = 0$) and is detected, the contract specifies 0 payment. This captures the idea that once manipulative behavior is exposed, the agent forfeits any contractual reward but faces no additional penalty.

For a contract $(F, \beta)$, an action $(a, b)$ is implementable if it satisfies the following two conditions.

**Incentive Compatibility (IC).** The agent does not gain by deviating:

$$U_{\text{agent}}(a_{k'}, b_{k'}) \leq U_{\text{agent}}(a_k, b_k) \quad \forall a_{k'} \in \mathcal{A}, \ b_{k'} \in \{0, 1\}.$$

**Individual Rationality (IR).** Participation yields non-negative utility:

$$U_{agent}(a_k, b_k) \geq 0.$$

Together, the IC and IR constraints ensure that the agent's optimal response to the principal's contract matches the principal's desired behavior.

Our mechanism proceeds in two stages:

**Step 1: Screening for high quality.** The principal designs a menu $F$ so that only high-quality agents $A_h$ find participation profitable. This rules out both $(A_l, 0)$ and $(A_l, 1)$. (Section 3.1)

**Step 2: Enforcing compliance.** An inspection probability $\beta$ is then chosen to deter non-compliance, eliminating $(A_h, 0)$ and leaving only $(A_h, 1)$. (Section 3.2)

### 3.1 STATISTICAL CONTRACTS

Below, we focus on the design of the statistical contract menu $F$. Let $\overline{\mathcal{F}}$ be the set of admissible payment functions. The principal selects a finite menu $F \subseteq \overline{\mathcal{F}}$. We assume a hypothesis-testing setup in which the action space $\mathcal{A}$ is partitioned into non-empty sets $A_l$ and $A_h$. The principal wishes to identify agents in $A_h$ while avoiding those in $A_l$. The principal's utility is

$$U_{\text{principal}} = r(a) - f(Z) - \beta d_k.$$

We require $U_{\text{principal}}$ to be non-negative and non-decreasing for $a \in A_h$, and non-positive and non-increasing for $a \in A_l$. If an agent declines the contract, the principal's payoff is zero. These

conclusions about menu design do not depend on the exact functional form of $U_{\text{principal}}$ or on the distribution $Q_a$ of evidence conditional on quality. Thus, effective menus $F$ can be constructed without precise knowledge of these quantities. An agent aims to maximize the expected payment,

$$f^{\text{opt}}(\cdot; a, b, F) = \arg\max_{f \in F}\{\mathbb{E}_Z[f(Z)]\}, \tag{2}$$

where the expectation is taken over the evidence distribution induced by the agent's effort and compliance choices. Here, $f^{\text{opt}}$ represents any element in $F$ that maximizes the agent's expected payment for behavior $(a, b)$, and it depends only on $(a, b)$ and menu $F$. The set $A_l$ represents actions whose quality fails to meet the principal's requirements. The principal seeks to prevent such actions. When designing the menu $F$, we focus on contracts that make it unprofitable for agents with low product quality to participate.

**Definition 1.** *If for all $a \in A_l$ and $f \in F$, we have $\mathbb{E}_a[f(Z)] \leq d_t$, then the statistical contract defined by the menu $F$ and cost $d_t$ is quality-inducing.*

An incentive-compatible contract aligns the principal's and agent's interests so that agents cannot profit from low-quality products when acting legally. To characterize such contracts, we introduce *e-values*, which come from hypothesis-testing theory.

**Definition 2.** *Let $Z \in \mathcal{Z}$ be a random variable. For any $a_0 \in A_l$ , if $\mathbb{E}_{Z \sim Q_{a_0}}[g(Z)] \leq 1$, then the function $g : \mathcal{Z} \to \mathbb{R}_{\geq 0}$ is called an e-value under the null hypothesis $a \in A_l$.*

To capture the central role of e-values, we emphasize their flexibility in designing incentive-compatible contracts under uncertainty. Unlike fixed-threshold p-values, e-values support adaptive, sequential hypothesis testing, making them ideal when evidence is gradually revealed or manipulable. In our model, the set $\mathcal{E}$ of e-values corresponds to scaled payment functions, and Proposition 1 shows that incentive compatibility requires selecting payments from $\mathcal{E}$. This duality prevents agents from benefiting without strong supporting evidence.

**Proposition 1.** *A menu $F$ produces an incentive-compatible statistical contract if and only if $F/d_t \in \mathcal{E}$, where $F/d_t = \{f/d_t : f \in F\}$.*

Thus, the payment functions in an incentive-compatible contract are in one-to-one correspondence with e-values, linking incentive alignment directly to hypothesis testing. While such contracts reduce strategic manipulation, they may not fully prevent non-compliant behavior. To strengthen compliance, Section 3.2 below introduces a mechanism that incorporates random inspections.

## 3.2 RANDOM INSPECTIONS

Having excluded low-quality actions, we now use inspections with probability $\beta$ to enforce legal compliance. By linking the agent's incentives to hypothesis testing, the mechanism ensures that agents can earn high rewards only when presenting strong evidence against the null hypothesis. This screening deters low-quality participants from joining the mechanism, but it cannot by itself prevent high-quality agents from taking illegal actions ($b = 0$). Therefore, the principal must conduct random and costly inspections to motivate compliance with safety regulations.

Before studying the probability $\beta$, we first present the principal's payoff and explain why the principal strictly prefers legal actions. Remember that $U_{\text{principal}}(a, b)$ is the principal's expected utility when the agent takes action $(a, b)$. For $b = 0$, the utility is $-\infty$, as negative consequences occur with a non-zero probability $\alpha$, resulting in catastrophic losses, so

$$U_{\text{principal}}(a, 0) = -\infty.$$

The mechanism must therefore be designed to eliminate such outcomes. For $b = 1$, the principal's payoff is given by:

$$U_{\text{principal}}(a, 1) = r(a) - T - \beta \cdot d_k, \tag{3}$$

where $r(a)$ is the reward from true quality, $T$ is payment to the agent, and $d_k$ is the cost of inspection.

**Assumption 2.** *For every agent, $T - c(a) \geq d_t$.*

This assumption guarantees that each agent has at least one implementable legal action.

Observing $T = f_j^{\mathrm{opt}}(Z)$, the principal chooses two key parameters: the payment function $f_j^{\mathrm{opt}} \in F$ and the inspection probability $\beta$. First, consider the case without inspections ($\beta = 0$). In this situation, the principal could try to design stronger menus to induce legal behavior, but this may fail.

**Lemma 1.** *If $\alpha < \frac{d_t}{r(a_n)}$, then no legal action is implementable without inspections.*

This demonstrates that when negative consequences are sufficiently rare, inspections are necessary to incentivize legal actions. To analyze the optimal inspection probability $\beta$, we introduce the concept of the payment ratio $\gamma$, which links the designed menu $F$ to the agent's behavior.

**Definition 3.** *Given an agent's action $(a, b)$ that yields reward $r(a)$ and a chosen payment function $f$, the payment ratio is*

$$\gamma = \frac{f(Z)}{r(a)}, \qquad \gamma \in [0, 1].$$

A higher payment ratio means the agent captures a larger share of the action's total benefit, strengthening incentives to choose high-quality, legal actions. Let $F = \{f_j\}_{j=1}^{l}$ be the menu provided by the principal. When the agent selects action $(a, b)$, they choose an optimal function $f_j$ from $F$. Because reward $r(a)$ is fixed, the agent's profit $f_j(Z)$ varies across the menu, and the corresponding set of possible payment ratios is $\{\gamma_j\}_{j=1}^{l}$.

We treat $\gamma$ as a piecewise-continuous variable on the interval $[0, 1]$ to facilitate subsequent analysis. Let $\gamma_j^{\mathrm{opt}}$ denote the optimal payment ratio that an agent can obtain from the set of payment ratios. Suppose an agent performing action $(a, b)$ selects the optimal payment function $f_j^{\mathrm{opt}}$ from the menu $F$, then the relationship between $\gamma_j^{\mathrm{opt}}$ and $f_j^{\mathrm{opt}}$ is given below.

**Proposition 2.** *When an agent taking action $(a, b)$ chooses the optimal payment function $f_j^{\mathrm{opt}}$ from the menu $F$, the corresponding payment ratio equals the optimal payment ratio:*

$$\gamma_j^{\mathrm{opt}} = \frac{f_j^{\mathrm{opt}}}{r(a)}. \tag{4}$$

Using this relationship, we next derive conditions for implementing legal actions and determining the associated inspection probability $\beta$.

**Proposition 3.** *Suppose Assumption 2 holds, when the agent selects the optimal payment function $f_j^{\mathrm{opt}}$ from the menu $F$, there exists $\beta(\gamma_j^{\mathrm{opt}})$, with $\gamma_j^{\mathrm{opt}} = f_j^{\mathrm{opt}}/r(a)$, such that for any intended inspection probability $\beta > \beta(\gamma_j^{\mathrm{opt}})$, the contract $(F, \beta)$ implements a legal action.*

When the menu $F$ is fixed, for any payment function $f_j$ chosen by the agent ($j \in \{1, ..., l\}$), any inspection probability $\beta > \beta(\gamma_j^{\mathrm{opt}})$ ensures that the agent act legally. Since $\beta$ affects only the principal's utility, we may, without loss of generality, set $\beta = \beta(\gamma_j^{\mathrm{opt}})$. We now analyze how the inspection probability $\beta(\gamma^{\mathrm{opt}})$ varies with $\gamma$.

**Lemma 2.** *Suppose Assumption 2 holds and recall the definition of $\beta(\gamma^{\mathrm{opt}})$ from Proposition 3. Consider $\gamma$ is piecewise continuous, then $\beta(\gamma^{\mathrm{opt}})$ is a decreasing function of $\gamma$. Moreover, whenever $\beta(\gamma^{\mathrm{opt}}) > 0$, it decreases strictly.*

Although inspections impose costs on the principal, they play a crucial role in motivating the agent to choose legal, high-quality actions that align with the principal's objectives. Rather than viewing inspections solely as a deterrent to misconduct, we formalize their role as a positive incentive: their presence strengthens a rational agent's motivation to comply.

**Theorem 1.** *Increasing the inspection probability $\beta$ strictly raises the relative payoff of compliance at every effort level.*

**Corollary 1.** *If the effort level induced by compliance exceeds that induced by non-compliance, then increasing $\beta$ can trigger a switch from non-compliance to compliance, and the equilibrium effort may jump upward at the switching threshold.*

Due to space constraints, the full proof is provided in the Appendix.

This result shows that a higher inspection probability $\beta$ encourages the agent to exert greater effort, reinforcing the principal's objective of aligning compliance with high-quality outcomes. Increasing $\beta$ raises the expected cost of violations and reduces the attractiveness of opportunistic behavior. As a result, the agent is incentivized to improve their work effort and comply with regulations rather than taking risks by violating them. The principal motivates the agent to choose high-quality actions through the payment function $f(Z)$, while the inspection probability $\beta$ shapes the design of that payment function. Together, the contract menu $F$ and the inspection probability $\beta$ form a unified incentive mechanism.

A concise outline of the overall procedure appears in Algorithm 1.

---

**Algorithm 1** Computing the Optimal Contract in a Single-Agent Environment

1: **Input:** Actions $a$ with benefits $r(a)$ and costs $c(a)$; Evidence $Z$; e-value set $E$; Adverse effect probability $\alpha$; Testing cost $d_t$.
2: **Output:** Optimal menu $F$; Optimal inspection probabilities $\{\beta^{\mathrm{opt}}\}$.
3: **for all** $g(Z) \in E$ **do**
4:     Define payment function: $f(Z) = d_t \cdot g(Z)$.
5: **end for**
6: Construct incentive-compatible menu: $F = \{f \mid f(Z) > d_t\}$.
7: **For** legal action $(a_{\mathrm{legal}}, 1)$ and illegal action $(a_{\mathrm{illegal}}, 0)$:

$$f_{\mathrm{legal}}^{\mathrm{opt}} = \arg\max_{f \in F} \mathbb{E}[f(Z(a_{legal}, 1))], \quad f_{\mathrm{illegal}}^{\mathrm{opt}} = \arg\max_{f \in F} \mathbb{E}[f(Z(a_{illegal}, 0))]$$

8: Define $\gamma^{\mathrm{opt}} = \frac{f^{\mathrm{opt}}}{r(a)}$.
9: **for all** $a$ **do**
10:     Solve $f_{legal}^{\mathrm{opt}}(Z) - c(a_{\mathrm{legal}}) - d_t = (1-\alpha)(1-\beta^{\mathrm{opt}})f_{\mathrm{illegal}}^{\mathrm{opt}}(Z) - c(a_{\mathrm{illegal}})$.
11:     Compute $\beta^{\mathrm{opt}} = 1 - \dfrac{f_{legal}^{\mathrm{opt}}(Z) - c(a_{\mathrm{legal}}) - d_t + c(a_{\mathrm{illegal}})}{(1-\alpha)f_{\mathrm{illegal}}^{\mathrm{opt}}(Z)}$.
12: **end for**
13: **Return:** Optimal menu $F$ and inspection probabilities $\{\beta^{\mathrm{opt}}\}$.

---

## 4 NUMERICAL EXPERIMENT

We conduct numerical experiments to evaluate our policy designs for agent selection and compliance. These experiments link the theoretical results to empirical evidence in a controlled setting. Due to space constraints, more experimental results are provided in the Appendix.

### 4.1 SETUP AND METRICS

We consider an asymmetric information environment with 10,000 agents. Each agent privately knows quality and compliance $(a, b)$, where $a \in [0, 1]$, and $b \in \{0, 1\}$. An agent decides whether to cheat; non-compliance as $b = 0$ and compliance as $b = 1$. The principal observes only a noisy scalar evidence $Z = Z_a + Z_b$, where $Z_a$ depends on effort and $Z_b \geq 0$ is a distortion present only when $b = 0$ (and equals 0 when $b = 1$). Throughout the simulations, we treat $Z_a$ as a numerical proxy for $a$.

To test robustness, we instantiate two evidence distributions for $Z_a$: i) a truncated normal on $[0, 1]$, representing a generic case, and ii) a chi-square distribution with $\mathrm{df} = 3$ rescaled to $[0, 1]$, representing a "lower-middle mass" scenario. Our theory is distribution-agnostic; these two families are used only for stress testing. Agent reward $r(a)$ and action cost $c(a)$ are modeled as linear in $a$, satisfying the required monotonicity so that higher quality yields higher reward and cost. We sweep key parameters on a grid: the adverse-event probability $\alpha$ takes 100 values in $[0.001, 0.1]$, and the testing cost $d_t$ takes 10 values in $[0.1, 1]$. Results in the main text are averaged across these configurations; detailed sensitivity analyses appear in the Appendix.

For statistical testing, we construct a family of threshold-based mappings as $e$-values. For each threshold $\tau$, define

$$f_\tau(Z) = d_t \cdot c_\tau \cdot \mathbf{1}\{Z \geq \tau\}, \qquad c_\tau^{-1} = \sup_{a \in A_l} \Pr(Z \geq \tau), \qquad \text{with } A_l = \{a \mid Z_a < \tau\},$$

so that $\sup_{a \in A_l} \mathbb{E}[f_\tau(Z) \mid a] \leq d_t$. The collection $\{f_\tau(Z) : \tau \in \mathcal{T}\}$ constitutes the $e$-value set used in our experiments. Algorithm 1 then produces the payment menu $F$ and inspection probabilities $\{\beta\}$ using the inputs $r(a)$, $c(a)$, $Z$, the $e$-value set $E$, $\alpha$ and $d_t$.

We evaluate three performance metrics: *Acceptance Rate* (the overall fraction of accepted agents), *HLR* (High-quality and Legal Rate), and *Illegal Action* (probability that an illegal action is accepted).

Policies compared are: *Joint(Ours)*, which combines a $Z$-based screen with follow-up inspections; *Tests-only*, which screens solely on $Z$ with no inspections; and *Inspections-only*, which relies entirely on random inspections.

## 4.2 BENCHMARK COMPARISON ACROSS METRICS

We first compare the three policy approaches in terms of Acceptance Rate (Acc), HLR, and Illegal Action (IA). Figure 1 below summarizes results.

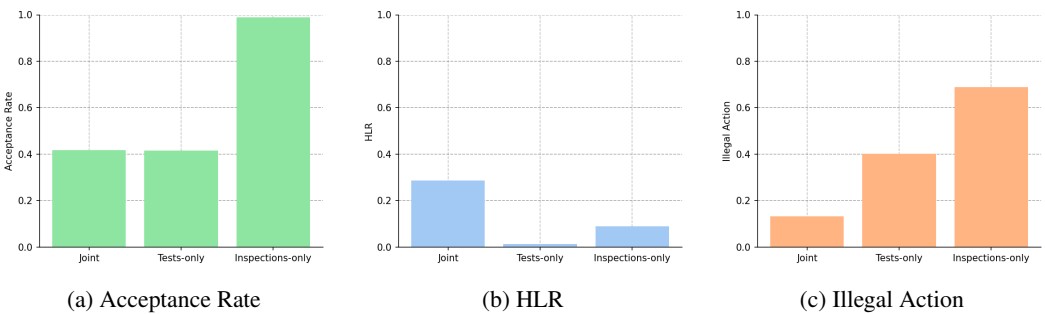

(a) Acceptance Rate      (b) HLR      (c) Illegal Action

Figure 1: Benchmark comparison across three metrics under Tests-only, Inspections-only, and Joint(Ours). The Joint policy achieves the highest HLR and the lowest illegal rate.

Figure 1a shows the *Acceptance Rate* for each method. As expected, Inspections-only admits nearly all agents because it imposes no test-based gate and hence rejects applicants upfront. By contrast, both Tests-only and Joint(Ours) apply a test-based screen and thus accept far fewer agents. Although Test-only and Joint have similar overall acceptance rates, the composition of their accepted sets differs: Joint accepts agents who, on average, meet a higher integrity standard (as reflected by higher HLR and lower IA in Figures 1b and 1c), whereas Test-only's nearly unfiltered intake contains many low-quality entrants.

Figure 1b reports the *HLR* achieved by Tests-only, Inspections-only, and Joint(Ours). Joint attains a substantially higher HLR than either alternative. Mechanistically, Joint first screens on $Z$ to discourage low-quality applicants and then uses inspections to deter cheating among those who pass. In Tests-only, reliance solely on $Z$ creates strong incentives for marginal agents to cheat to meet the threshold, which reduces the share of genuinely high-quality, compliant selections. In Inspections-only, the absence of a front-end test admits many low-quality agents, diluting the fraction of high-quality compliant agents. Thus, combining screening and inspections is critical for raising HLR.

Figure 1c shows the *Illegal Action* (IA) for each policy, defined as the probability that an illegal action is accepted. Consistent with the previous results, Joint maintains the lowest IA, Tests-only yields higher IA, and Inspections-only yields the highest IA. For Joint and Tests-only, the accounting identity $\text{Acc} \approx \text{HLR} + \text{IA}$ holds approximately. Thus, among agents accepted by Joint, nearly $70\%$ are HLR, with the remainder primarily illegal actions. By contrast, Tests-only admits many illegal actions because it lacks a post-test deterrence mechanism. For Inspections-only, $\text{Acc} > \text{HLR} + \text{IA}$; the gap corresponds to low-quality but legal entrants who are admitted due to the lack of a test screen. Overall, Joint suppresses illegal actions most effectively by pairing evidence-based screening with calibrated inspections.

## 4.3 ROBUSTNESS ACROSS DISTRIBUTIONS

To evaluate the generality of our findings, we test how each policy performs under different data distributions. Specifically, we re-run the evaluation of HLR and Illegal Action using two distinct distributions for the performance signal $Z_a$: a standard normal distribution and a chi-square distribution. Figure 2a reports the HLR of Tests-only, Inspections-only, and Joint(Ours) policies under these distributions, and Figure 2b shows the corresponding illegal action results.

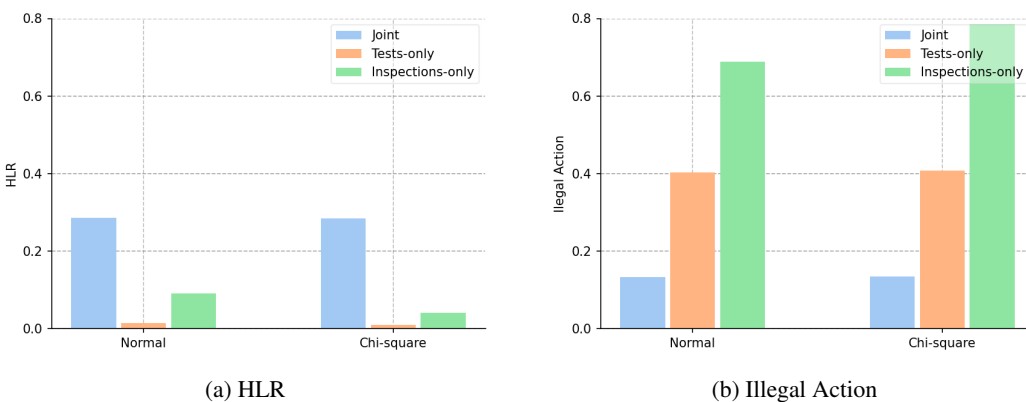

(a) HLR            (b) Illegal Action

Figure 2: Robustness of HLR and Illegal Action under normal and chi-square distributions. Joint design is robust to the choice of the evidence distribution, whereas the other methods are sensitive.

The results demonstrate that the Joint(Ours) design is robust to the choice of the evidence distribution, whereas the other methods are sensitive. This stability indicates that Joint's ability to select high–quality compliant agents and suppress illegal actions does not materially degrade when the underlying data distribution shifts.

In Tests-only, the heavier-tailed or more skewed evidence regime produces more borderline and noisy scores; without any post-test deterrence, marginal agents face strong incentives to cheat to clear the threshold, which lowers the share of genuinely high-quality legal acceptances and nudges the illegal fraction upward. For Inspections-only, the deterioration is substantial: HLR falls by more than half, and IA increases sharply. Because there is no front-end test, a shift toward lower–middle quality admits many more weak agents. Given inspection costs, many of these entrants still choose to violate, so the fraction of high-quality legal selections shrinks while illegal actions rise.

Across the two evidence families, Joint(Ours) preserves a high HLR and a moderate IA with negligible drift, whereas Tests-only and Inspections-only exhibit large swings—especially the latter under the chi-square regime. This confirms that combining evidence-based screening with calibrated inspections yields distributionally robust performance, unlike relying on either component alone.

## 5 CONCLUSION

In our research, we presented an incentive mechanism that integrates statistical hypothesis testing with strategic random inspections to address information asymmetry in principal–agent settings. Our design uses incentive-compatible statistical contracts and e-values to set payments, while a calibrated inspection policy deters fraud and low-effort behavior. Theoretical analysis shows that the two components reinforce each other: statistical evidence guides rewards, and inspections sustain truthful reporting. Extensive simulations confirm that the mechanism achieves both high-quality effort and strong compliance at competitive cost.

This work highlights how statistical inference and contract theory can be combined to create practical policies for settings such as clinical trials, online platforms, and recruitment. Future research could extend our model by incorporating dynamic agent behaviors, multi-agent interactions, and adaptive inspection strategies to enhance practical applicability.

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

# A APPENDIX

## A.1 USE OF LARGE LANGUAGE MODELS

Large language models (LLMs) were employed solely as a general-purpose writing aid. Specifically, we used an LLM to check grammar, improve sentence clarity, and refine word choice in the manuscript. The model did not contribute to the research ideas, study design, analysis, or interpretation of results. All conceptual and scientific content remains entirely the responsibility of the authors.

## A.2 ADDITIONAL EXPERIMENTAL DETAILS AND RESULTS

This appendix complements the numerical experiments in the main text by (i) briefly restating the data–generation pipeline and parameterization used throughout, and (ii) presenting additional visualizations that clarify how the proposed mechanism behaves under different signal regimes and design choices.

### A.2.1 EXPERIMENTAL SETUP RESTATE

We consider asymmetric information: each agent privately knows $(a, b)$ with $a \in [0, 1]$ and $b \in \{0, 1\}$ (noncompliance $b = 0$, compliance $b = 1$) and decides whether to cheat as a function of $a$. The principal observes only $Z = Z_a + Z_b$, where $Z_a$ is effort–dependent and $Z_b \geq 0$ appears only if $b = 0$ (and $Z_b = 0$ when $b = 1$); in simulations, we use $Z_a$ as a numerical proxy for $a$.

We probe robustness with two evidence families for $Z_a$: (i) a truncated normal on $[0, 1]$ (general case), and (ii) a chi-square with $\mathrm{df} = 3$ rescaled to $[0, 1]$ (mass in the lower–middle range). Theoretical guarantees are distribution-agnostic in $Q_a$; the dual instantiation serves to stress-test empirical performance. Agent payoff and cost follow linear schedules $r(a)$ and $c(a)$ satisfying monotonicity. We sweep $\alpha \in [0.001, 0.1]$ (100 values) and $d_t \in \{0.1, 0.2, \ldots, 1.0\}$ (10 values); main-text results report averages across this grid, with detailed sensitivities in the appendix.

For statistical testing, we build a threshold–indexed e–value menu: for $\tau \in \mathcal{T}$, $f_\tau(Z) = d_t c_\tau \mathbf{1}\{Z \geq \tau\}$ with $c_\tau^{-1} = \sup_{a \in A_l} \Pr(Z \geq \tau)$ and $A_l = \{a : Z_a < \tau\}$, so that $\sup_{a \in A_l} \mathbb{E}[f_\tau(Z) \mid a] \leq d_t$. Finally, we feed $(r(a), c(a), Z, E, \alpha, d_t)$ into Algorithm 1 (see Section 3.2) to compute the payment menu $F$ and inspection policy $\{\beta\}$ by solving the incentive–compatibility constraints.

### A.2.2 INSPECTION PROBABILITY AS A FUNCTION OF OBSERVED EVIDENCE

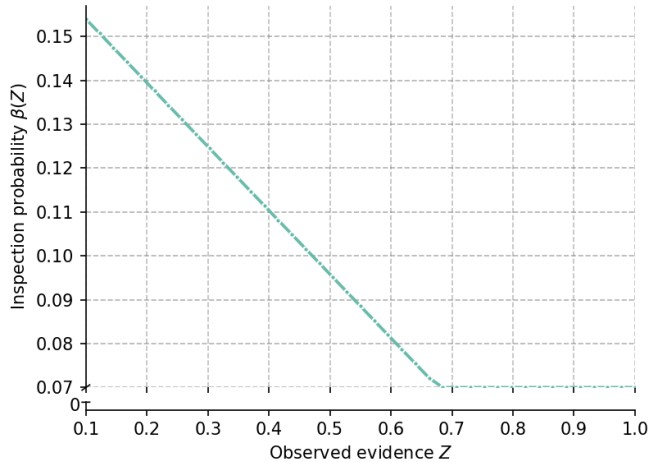

Figure 3: Evidence–dependent inspection under the joint design. The policy is monotone decreasing in $Z$ with a small positive floor at high $Z$, balancing efficiency (fewer costly audits for clearly high performers) and prudence (no agent is fully exempt from monitoring).

Inspection intensity declines with stronger test evidence, while a nonzero lower bound is retained even at the top end. This matches the mechanism's intuition: strong signals indicate likely high–quality and compliant agents (hence fewer audits), whereas borderline cases are inspected more often. The positive floor preserves universal deterrence by ensuring that no agent is fully exempt from oversight.

### A.2.3 EFFECT OF CONTRACT MENU SIZE ON HLR

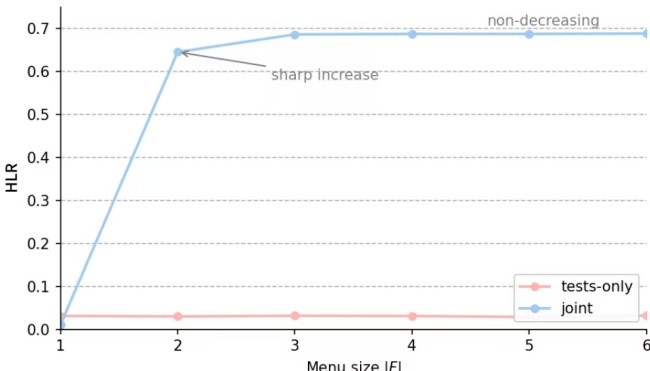

Figure 4: HLR versus menu size $|F|$. A large gain from $|F| = 1$ to $|F| = 2$ is followed by diminishing returns; small menus with 2–3 well–chosen options achieve performance close to saturation.

Adding a second option captures most of the personalization benefit; beyond three options, the curve is nearly flat. Practically, the principal does not need a large, complex menu to obtain near–optimal HLR.

### A.2.4 COMPLIANCE PROBABILITY AS A FUNCTION OF QUALITY SIGNAL

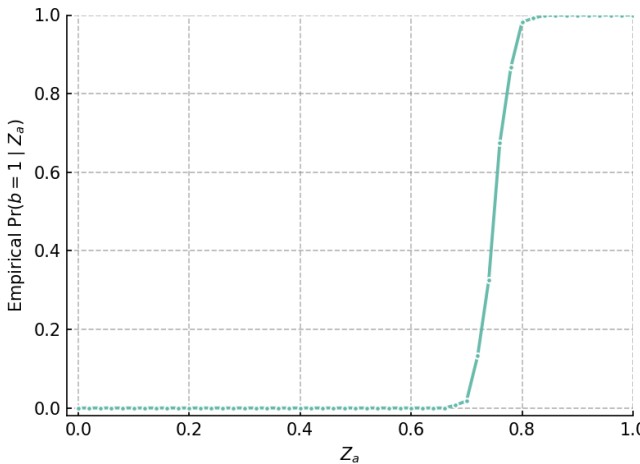

Figure 5: Empirical $\Pr(b = 1 \mid Z_a)$ rises sharply with $Z_a$: low–quality agents tend to cheat, whereas high–quality agents almost always comply.

The logistic relationship reflects the intuitive trade–off: weak agents face stronger pressure to cheat to appear qualified, while strong agents can meet requirements legally and thus prefer compliance.

### A.2.5 SENSITIVITY OF ILLEGAL ACTION

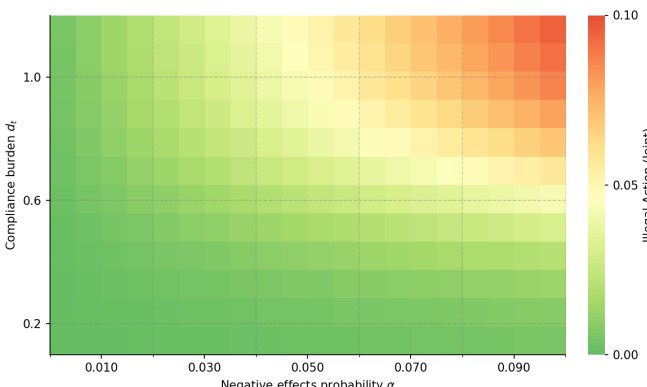

Figure 6: Heatmap of the Illegal Action metric versus punishment $\alpha$ and testing cost $d_t$. The Illegal Action increases with both $\alpha$ and $d_t$ across the grid.

The heatmap shows a clear upward trend along both axes: larger $\alpha$ or larger $d_t$ is associated with higher Illegal Action.

### A.3 PROOFS

### A.3.1 PROOF OF PROPOSITION 2.4.

*Proof.* According to Definition 2, the e-value $g(Z)$ satisfied

$$\mathbb{E}_{Z \sim Q_{a_0}}[g(Z)] \leq 1, \forall a_0 \in A_l,$$

where the e-value is a statistical tool based on hypothesis testing. It measures the degree to which data refuses the null hypothesis $a_0 \in A_l$. A larger value indicates stronger evidence against the null hypothesis.

First, we prove the sufficiency. Suppose $F/d_t$ belongs to the set of e-values $\mathcal{E}$, which means that for all $f \in F$,

$$\mathbb{E}_{Z \sim Q_{a_0}}[\frac{f(Z)}{d_t}] \leq 1, \forall a_0 \in A_l,$$

which is equivalent to:

$$\mathbb{E}_{Z \sim Q_{a_0}}[f(Z)] \leq d_t, \forall a_0 \in A_l.$$

Referring to Definition 1 of an incentive-compatible statistical contract, if for all $a \in A_l$ and all $f \in F$, the expected value $\mathbb{E}_a[f(Z)] \leq d_t$, then the statistical contract is incentive-compatible. Hence, if $F/d_t \in \mathcal{E}$, we can directly infer that $F$ generates an incentive-compatible statistical contract.

Similarly, for necessity, if $F$ generates an incentive-compatible statistical contract, then for all $a \in A_l$ and all $f \in F$, we have:

$$\mathbb{E}_a[f(Z)] \leq d_t.$$

Dividing both sides by $d_t$(since $d_t > 0$), we obtain:

$$\mathbb{E}_a[\frac{f(Z)}{d_t}] \leq 1,$$

which is precisely the definition of an e-value. That is, $\frac{f(Z)}{d_t} \in \mathcal{E}$. Therefore, for all $f \in F$, we conclude that $\frac{F}{d_t} \in \mathcal{E}$. □

### A.3.2 PROOF OF LEMMA 2.6.

*Proof.* We now prove that when the probability $\alpha$ of adverse effects occurring is too low, all legal actions $(a, 1)$ fail to satisfy the incentive compatibility condition, requiring stronger inspections $(\beta > 0)$ to enforce compliance.

- Case 1: The agent chooses a legal action $(b = 1)$.

  If the agent selects a legal action (i.e., complies with safety regulations), the utility is:

  $$U_{agent}(a, 1) = f(Z) - c(a) - d_t.$$

- Case 2: The agent chooses an illegal action $(b = 0)$ If the agent chooses an illegal action $(b = 0)$, and is not discovered, the utility is:

  $$U_{agent}(a, 0) = (1 - \alpha)f(Z) - c(a).$$

To ensure that legal actions are the optimal choice, we require:

$$U_{agent}(a, 1) \geq U_{agent}(a, 0),$$

which simplifies to:

$$f(Z) - c(a) - d_t \geq (1 - \alpha)f(Z) - c(a).$$

Rearranging:

$$\alpha \geq \frac{d_t}{f(Z)}.$$

Since the payment function $f(Z)$ does not exceed $r_n$, taking the maximum $f(Z) = r_n$, we obtain $\alpha \geq \frac{d_t}{r_n}$.

If we obtain $\alpha < \frac{d_t}{r_n}$, which means that for all $f(Z) \leq r_n$, the condition $\alpha f(Z) \geq d_t$ cannot be satisfied. This implies that the expected utility $U_{agent}(a, 0)$ of the agent choosing an illegal action is always greater than the utility $U_{agent}(a, 1)$ of choosing a legal action. Consequently, all legal actions become unimplementable. Thus, in the absence of inspections$(\beta = 0)$, the agent will not choose legal actions, leading to the failure of the compliance mechanism. Additional inspections$(\beta > 0)$ are required to enforce legal actions. $\square$

### A.3.3 PROOF OF PROPOSITION 2.8.

*Proof.* We note that when the agent's action $(a, b)$ is determined, the menu $F$ provided by the principal and the reward $r_i$ from the agent's action are also fixed. Since the agent selects the payment function $f_i^{\mathrm{opt}}$ that maximizes their earnings, in the equation $\frac{f(Z)}{r}$, the denominator is fixed. The numerator takes the maximum value, meaning that the value of $\gamma$ is maximized, i.e., equation 4. $\square$

### A.3.4 PROOF OF PROPOSITION 2.9.

*Proof.* First, note that the agent's action is determined based on the menu provided by the principal, and the menu designed by the principal determines the agent's implemented actions. In other words, the goal is to obtain actions $(a_i, 1), i \in \{n\}$ that can be implemented as legal actions in the currently set menu $F$. Given the menu $F$, we need to eliminate the possibility that a rational agent may choose an illegal action, i.e., violating behavior $(a_w, 0), w \in \{n\}$. Recall that the agent's utility from such an action is:

$$(1 - \alpha)(1 - \beta)f_w(Z) - c_w.$$

Thus, the maximum utility that an agent can obtain from selecting the optimal payment function after implementing an illegal action is given by:

$$\max_w((1 - \alpha)(1 - \beta)f_w(Z) - c_w))$$

$$= (1 - \alpha)(1 - \beta)f_w^{\mathrm{opt}}(Z) - c_w.$$

An agent makes different choices under honest and dishonest circumstances. The optimal payment functions in these cases are denoted as $f_i^{\mathrm{opt}}$ and $f_w^{\mathrm{opt}}$, respectively. According to the incentive compatibility (IC) constraint, the following must be satisfied:

$$f_i^{\mathrm{opt}} - c(a) - d_t \geq (1-\alpha)(1-\beta)f_w^{\mathrm{opt}}(Z) - c_w. \tag{5}$$

Note that the left-hand side is non-positive and the payment function $f$ is an increasing function, we can conclude that there exists $\beta(\gamma_j^{\mathrm{opt}}) \geq 0$ such that when $\beta > \beta(\gamma_j^{\mathrm{opt}})$, equation (5) holds. $\qquad\square$

### A.3.5 PROOF OF LEMMA 2.10.

*Proof.* This lemma states that when the payment ratio $\gamma^{\mathrm{opt}}$ is large, agents have a stronger incentive to comply, allowing the principal to reduce the required inspection probability $\beta(\gamma^{\mathrm{opt}})$. Conversely, when $\gamma^{\mathrm{opt}}$ is small, agents have a stronger incentive to cheat, requiring higher inspection rates to enforce compliance. Recall the incentive compatibility condition:

$$f_i^{\mathrm{opt}}(Z) - c(a) - d_t \leq (1-\alpha)(1-\beta)f_w^{\mathrm{opt}}(Z) - c_w.$$

Using Proposition 2.9, we obtain the minimum value of $\beta$:

$$\beta(\gamma^{\mathrm{opt}}) = 1 - \frac{f_i^{\mathrm{opt}}(Z) - c(a) - d_t + c_w}{(1-\alpha)f_w^{\mathrm{opt}}(Z)}.$$

Since $\gamma^{\mathrm{opt}} = \frac{f_i^{\mathrm{opt}}(Z)}{r}$, we rewrite $\beta(\gamma^{\mathrm{opt}})$ as:

$$\beta(\gamma^{\mathrm{opt}}) = 1 - \frac{\gamma^{\mathrm{opt}}r - c(a) - d_t + c_w}{(1-\alpha)f_w^{\mathrm{opt}}(Z)}.$$

Taking the derivative of $\beta(\gamma^{\mathrm{opt}})$ with respect to $\gamma^{\mathrm{opt}}$:

$$\frac{d\beta}{d\gamma^{\mathrm{opt}}} = -\frac{r}{(1-\alpha)f_w^{\mathrm{opt}}(Z)}.$$

Since $r$ and $f_w^{\mathrm{opt}}(Z)$ are both positive and $1-\alpha > 0$, we have $\frac{d\beta}{d\gamma^{\mathrm{opt}}} < 0$, meaning that the required inspection probability $\beta$ decreases as the payment ratio $\gamma^{\mathrm{opt}}$ increases, proving that $\beta(\gamma^{\mathrm{opt}})$ is a decreasing function. When $\beta(\gamma^{\mathrm{opt}}) > 0$, we have $f_w^{\mathrm{opt}}(Z) > 0$, therefore :

$$\frac{d\beta}{d\gamma^{\mathrm{opt}}} = -\frac{r}{(1-\alpha)f_w^{\mathrm{opt}}(Z)} < 0.$$

Since it is strictly less than zero, this indicates that $\beta(\gamma^{\mathrm{opt}})$ is strictly decreasing. $\qquad\square$

### A.3.6 PROOF OF THEOREM 1

*Proof.* Fix an effort level $a$. Let the expected payment under evidence distribution $P_a$ be positive, that is $\mathbb{E}_{P_a}[f(Z)] > 0$, and $0 \leq \alpha < 1$. Define the agent's expected utilities under compliance and non-compliance by

$$\begin{aligned} U_1(a) &= \mathbb{E}_{P_a}[f(Z)] - c(a) - d_t, \\ U_0(a,\beta) &= (1-\alpha)(1-\beta)\,\mathbb{E}_{P_a}[f(Z)] - c(a). \end{aligned}$$

where $c(a)$ is twice differentiable with $c'(a) > 0$ and $c''(a) > 0$, and $d_t \geq 0$ is the direct cost of compliance. Define the utility gap

$$\Delta(a,\beta) := U_1(a) - U_0(a,\beta).$$

We compute the utility gap explicitly. By definition, we have

$$\begin{aligned} \Delta(a,\beta) &= U_1(a) - U_0(a,\beta) \\ &= \mathbb{E}_{P_a}[f(Z)] - c(a) - d_t - \big((1-\alpha)(1-\beta)\,\mathbb{E}_{P_a}[f(Z)] - c(a)\big) \\ &= \big(1 - (1-\alpha)(1-\beta)\big)\mathbb{E}_{P_a}[f(Z)] - d_t. \end{aligned}$$

Simplifying the coefficient yields

$$1 - (1 - \alpha)(1 - \beta) = \alpha + \beta - \alpha\beta.$$

Differentiate $\Delta(a, \beta)$ with respect to $\beta$ while holding $a$ fixed. The derivative is

$$\frac{\partial \Delta(a, \beta)}{\partial \beta} = (1 - \alpha)\, \mathbb{E}_{P_a}[f(Z)].$$

Under the stated assumptions we have $1 - \alpha > 0$ and $\mathbb{E}_{P_a}[f(Z)] > 0$, so the derivative is strictly positive. This proves that for any fixed effort level $a$, the relative payoff of compliance compared to non-compliance increases strictly as the inspection probability $\beta$ increases. $\qquad\square$

### A.3.7 PROOF OF COROLLARY 1

*Proof.* Define the agent's expected utilities under compliance and non-compliance by

$$
\begin{aligned}
U_1(a) &= \mathbb{E}_{P_a}[f(Z)] - c(a) - d_t, \\
U_0(a, \beta) &= (1 - \alpha)(1 - \beta)\, \mathbb{E}_{P_a}[f(Z)] - c(a).
\end{aligned}
$$

Define the utility gap

$$\Delta(a, \beta) := U_1(a) - U_0(a, \beta).$$

Let $a_1^*$ denote the effort that maximizes $U_1(a)$ and let $a_0^*$ denote the effort that maximizes $U_0(a, \beta)$ for a given $\beta$. As shown in Theorem 1, $\Delta(a, \beta)$ is strictly increasing in $\beta$. If $a_1^* > a_0^*$ and the agent chooses between compliance and non-compliance by comparing the maximized utilities, then there exists a threshold $\beta^\dagger$ such that

$$\max_a U_1(a) > \max_a U_0(a, \beta), \quad \forall \beta > \beta^\dagger.$$

For $\beta > \beta^\dagger$ the agent prefers compliance, so the equilibrium effort jumps from $a_0^*$ to $a_1^*$. Because this switch is discrete, the equilibrium effort as a function of $\beta$ can be discontinuous at $\beta^\dagger$. $\qquad\square$