# OpenReview forum: "Designing Contracts for Effort and Regulatory Compliance"
_ICLR.cc/2026/Conference — Submitted to ICLR 2026_

### Official Review · Reviewer_8eGk · 2025-10-25

**Soundness:** 2
**Presentation:** 2
**Contribution:** 2
**Rating:** 2
**Confidence:** 3

**Summary:**

In designing contracts, two issues are important: (1) encouraging the agent to conduct "high-quality" actions so that better outcomes are derived. (2) ensuring that no misreporting or manipulation of the outcome occurs.
This paper studies the mechanism design problem facing the above two kinds of incentives.
An optimal contract is provided.

**Strengths:**

+ The problem is well-motivated in the sense that outcome misreporting problems are prevalent in real life, but are less studied.

**Weaknesses:**

- However, I am not sure whether the author(s) model this problem in the right way.
Naturally, an agent should decide whether to comply after the outcome is revealed, yet the paper models that the effort action and the compliance action are chosen simultaneously, which seems unintuitive to me.
- I am not clear on the main result(s) of the paper. The author(s) claim that Section 3.1 already rules out the low quality action (the authors do not state how to divide low and high quality actions), but I feel that this condition only eliminates the pair of low quality action plus non-compliance. The main theorem also does not seem to provide the main message.

In all, I highly encourage the authors to carefully reorganize the materials and results for a clearer view.

**Questions:**

Could you briefly summarize your results and explain how you derive them more intuitively?

---

### Official Review · Reviewer_aVVv · 2025-11-01

**Soundness:** 3
**Presentation:** 3
**Contribution:** 2
**Rating:** 2
**Confidence:** 3

**Summary:**

The paper studies contract design that mixes evidence-based payments (built from e-values) with random inspections to incentivize agents toward high quality and legal behavior. Simulations show the joint design beats tests-only and inspections-only on "high-quality and legal rate" and on lowering illegal actions.

**Strengths:**

* The paper shows a connection between incentive-compatible contracts and e-values, which seems a clean connection between contract theory and hypothesis testing. It would be great if the authors could provide more background information on e-values for its significance in contract theory.

* The optimal contract is simple.

**Weaknesses:**

* The technique in the paper doesn't seem to be novel.

* I'm concerned with the novelty of the paper. There is a huge literature on both contract design and costly verification. The paper only discusses a few of them. I don't find the related work section very helpful. For example, the related work section on mechanism design names several papers without summarizing the difference (only a summary of the current paper is provided). I can list some papers that seem relevant, but the paper should include a more extensive discussion. e.g.

Townsend, R. M. (1979). Optimal contracts and competitive markets with costly state verification. Journal of Economic theory, 21(2), 265-293.

Lacker, J. M., & Weinberg, J. A. (1989). Optimal contracts under costly state falsification. Journal of Political Economy, 97(6), 1345-1363.

**Questions:**

It would be great if the authors could provide clarifications on novelty of both modeling and results.

---

### Official Review · Reviewer_3ZXo · 2025-11-01

**Soundness:** 2
**Presentation:** 1
**Contribution:** 2
**Rating:** 2
**Confidence:** 3

**Summary:**

The paper addresses a principal–agent setting in which a principal must incentivize both effort (to ensure high quality) and compliance (to prevent illegal or manipulative behavior). The paper introduces the model and its theoretical background, characterizes its features, and describes an algorithm to compute the optimal contract. The paper concludes with a few numerical experiments.

**Strengths:**

I quite like the model and motivations. Not only does this model seem to be timely and relevant, but it is also interesting from a theoretical perspective.

**Weaknesses:**

The writing quality is fairly poor for an ML conference. Since ICLR is an ML conference, the writing should be modified accordingly. I feel the paper style aligns more with a pure Econ paper (and I go so far as to say this paper is probably better suited to something like EC).
At the middle of section 3 I start to fail to see in which direction is the paper is going. Do you want to characterize the optimal menus? Or answering questions on the Econ implications? It is not clear.
Moreover, the proofs are mainly direct algebraic manipulations and fairly straightforward.

Minor/Suggestions:
* The 2004 Vioxx is not clear to me.
* Definition of $Z = Z(a, b) = Z_a + Z_b$ needs refinement. For example, $Z_b$ is said to be $\ge 0$ redundantly (both in general and when $b=0$). Maybe consider using an indicator?

**Questions:**

* In line 138, it seems that the agent first produces an outcome (or evidence, as called in this paper) and then selects a payment scheme. This seems odd; in fact, the ordering in the box at line 145 is reversed. What’s going on here?
* If there is a positive prob $\alpha$ of a catastrophic outcome occurring, with a reward of $-\infty$ to the principal when $b=0$, then the principal should make $b=1$ a dominant strategy (since the prob of illegal action being undetected is always $>0$? Am I reading this correctly? If so, this should probably be mentioned
* Line 154 / 138. What is $f_j$, what is $l$? Why am I offering a menu and not a single contract? Does the revelation principle not hold?
* Line 193, what is the quantifier on $k$?
* Line 213: where I see $-infinity$ in case of a non-detected illegal action?
* $U_{principal}$ is not a function of anything?

---

### Official Review · Reviewer_KQeG · 2025-11-05

**Soundness:** 2
**Presentation:** 2
**Contribution:** 2
**Rating:** 2
**Confidence:** 4

**Summary:**

This paper considers a contract design problem where principals want to incentivize agents to both exert high effort and truthfully report the outcome, given that the agent's actions are unobservable and they might fake evidence to look good. The key idea is to combine statistical hypothesis testing (for screening quality) with random inspections (for deterring illegality). The proposed mechanism, built around e-values from hypothesis-testing theory, links payments to evidence strength and introduces a minimum inspection probability to maintain incentive compatibility. Theoretical analysis establishes when inspections are necessary, derives their optimal probabilities, and proves that raising inspection frequency strictly increases compliance incentives. Extensive simulations show that the proposed mechanism (tests + inspections) outperforms “tests-only” and “inspections-only” baselines.

**Strengths:**

The paper studies the problem from both a theoretical and empirical approach.

**Weaknesses:**

- The paper has very limited results on this problem given the prior work.
- The paper also relies on some restrictive assumptions that lack practical motivation
- The formatting of citation (\citet, \citep) is not consistent

**Questions:**

The paper hints on the complementarity between statistical contracts and inspections. Could the authors formalize or quantify the marginal value of adding one component when the other already exists?

---

### Meta-Review · Area_Chair_PZXq · 2026-01-05

**Summary:**

Rejection was the unanimous decision due to marginal/insufficient results, lack of novelty and/or technical depth. The authors didn't submit a rebuttal.

**Reviewer Concerns:**

See above

**Reviewer Scores:**

NA

---

### Decision · Program_Chairs · 2026-01-26

Reject